# cryo-EM structures of the *E. coli* replicative DNA polymerase reveal its dynamic interactions with the DNA sliding clamp, exonuclease and τ

**Rafael Fernandez-Leiro[†], Julian Conrad[†], Sjors HW Scheres\*, Meindert H Lamers\***

MRC Laboratory of Molecular Biology, Cambridge, United Kingdom

**Abstract** The replicative DNA polymerase PolIIIα from *Escherichia coli* is a uniquely fast and processive enzyme. For its activity it relies on the DNA sliding clamp β, the proofreading exonuclease ε and the C-terminal domain of the clamp loader subunit τ. Due to the dynamic nature of the four-protein complex it has long been refractory to structural characterization. Here we present the 8 Å resolution cryo-electron microscopy structures of DNA-bound and DNA-free states of the PolIII-clamp-exonuclease-$\tau_c$ complex. The structures show how the polymerase is tethered to the DNA through multiple contacts with the clamp and exonuclease. A novel contact between the polymerase and clamp is made in the DNA bound state, facilitated by a large movement of the polymerase tail domain and $\tau_c$. These structures provide crucial insights into the organization of the catalytic core of the replisome and form an important step towards determining the structure of the complete holoenzyme.

**\*For correspondence:** scheres@mrc-lmb.cam.ac.uk (SHS); mlamers@mrc-lmb.cam.ac.uk (MHL)

[†]These authors contributed equally to this work

## Introduction

In *Escherichia coli*, DNA replication is highly efficient with speeds of up 600–1000 nucleotides per second (*Mok and Marians, 1987*; *Mcinerney et al., 2007*), >100,000 basepairs (bp) synthesized per binding event (*Yao et al., 2009*), and an error rate of ~1 per million (*Bloom et al., 1997*). Importantly, DNA replication is greatly complicated by the antiparallel orientation of the two DNA strands that need to be replicated simultaneously. To do so, DNA replication is performed by a large multi-protein complex termed the DNA polymerase III holoenzyme that synthesizes the leading strand in a continuous manner, while the lagging strand is synthesized in short fragments of ~1000 bp. The holoenzyme is composed of 10 subunits (α, β, ε, θ, δ, δ', γ, τ, χ, ψ), that together with the helicase DnaB and the RNA primase DnaG form the replisome with a combined molecular weight of 1 MDa. The replisome can be divided into three functional subcomplexes that together catalyze a series of events. The helicase DnaB separates the two DNA strands (*Mok and Marians, 1987*) and transiently associates with the RNA primase DnaG that synthesizes short RNA primers required for DNA synthesis at the lagging strand (*Wu et al., 1992*). The clamp loader subcomplex (δ, δ', γ, τ, χ, ψ) loads the DNA sliding clamp β, the processivity factor for the DNA polymerase, onto the DNA (*Stukenberg et al., 1991*). It furthermore connects the leading and lagging strand polymerases via its τ subunits (*McHenry, 1982*; *Onrust et al., 1995*). Finally, DNA synthesis is performed by the polymerase subcomplex that contains the DNA polymerase III α (PolIIIα), the DNA sliding clamp β, the proofreading exonuclease ε, and the C-terminal domain of the clamp loader subunit τ. The activity of PolIIIα is poor in isolation (*Maki and Kornberg, 1985*) and is greatly enhanced by its associated proteins. For error-free DNA synthesis the polymerase relies on the exonuclease ε that removes any misincorporated bases and decreases the error rate of DNA replication by 1–2 orders of

**eLife digest** DNA replication is complicated because the two strands that form its "double helix" structure run in opposite directions and need to be replicated at the same time. One of the new strands, the leading strand, is built continuously. While the other strand, called the lagging strand, is made in stretches that are about 1000-times shorter and run in the opposite direction from the leading strand. This means that the enzyme that builds the new strands of DNA (called DNA polymerase) must be repeatedly released and repositioned when it builds the lagging strand, however it is not fully understood how this achieved.

Fernandez-Leiro, Conrad et al. have now used a technique called cryo-electron microscopy to reveal the three-dimensional structure of a DNA polymerase from a bacterium called *Escherichia coli* complete with other associated factors and a DNA molecule. These factors include: the "sliding clamp" that allows the polymerase to slide along the DNA; the "proofreading exonuclease" that removes mistakes in the newly built DNA strand, and the "processivity switch Tau" that is needed for the repeated release and repositioning of the polymerase at the lagging strand. These structures show how the polymerase is bound to the DNA by multiple interactions with the sliding clamp and exonuclease.

Fernandez-Leiro, Conrad et al. also solved the structure of the same proteins but without the DNA molecule. This revealed a large structural change between the DNA-bound and DNA-free states, which provides some clues as to how the polymerase can be quickly released from the DNA during the repeated cycles of DNA synthesis at the lagging strand. Further research is now needed to uncover what signals trigger this release of the DNA polymerase.

magnitude (*Scheuermann et al., 1983*; *Lancy et al., 1989*). In addition, the exonuclease strengthens the interactions between the polymerase and clamp as it binds both proteins simultaneously (*Toste Rêgo et al., 2013*; *Jergic et al., 2013*). For processivity, PolIII$\alpha$ binds to the DNA sliding clamp ($\beta$ subunit) (*Stukenberg et al., 1991*). At the leading strand this interaction is stable and results in DNA segments of >100.000 bp synthesized per binding event (*Yao et al., 2009*). At the lagging strand in contrast, DNA synthesis is discontinuous, with an averaged length of 1000 bp synthesized per fragment, depending on the frequency of the RNA primase activity (*Wu, Zechner and Marians, 1992*). This therefore requires repeated binding and release of the polymerase and clamp. Finally, the C-terminal domain of $\tau$ ($\tau_c$) acts as a 'processivity switch' for the polymerase to enable repeated binding and release at the lagging strand (*Leu et al., 2003*; *Georgescu et al., 2009*). How this tetrameric complex of PolIII$\alpha$-clamp-exonuclease-$\tau_c$ assembles and how it is repeatedly loaded and released during lagging strand synthesis is poorly understood.

The structures of the helicase-primase subcomplex (*Bailey et al., 2007*; *Wang et al., 2008*) and the clamp loader subcomplex (*Jeruzalmi et al., 2001*; *Simonetta et al., 2009*) have been known for some time. The structure of the PolIII$\alpha$-clamp-exonuclease-$\tau_c$ complex on the other hand has remained elusive due to its dynamic nature that forms a significant hurdle for structure determination. To overcome this, we have used a combination of site directed mutagenesis and computational classification of different structural states to determine the cryo-EM structures of the complex in both a DNA-bound and a DNA-free state to 8 Å resolution. The well defined features of the cryo-EM maps enable the unambiguous fitting of the crystal structures of the individual proteins, revealing the unique interactions between the four proteins and DNA. In the DNA-bound complex, the polymerase is tethered to the DNA through multiple contacts with the clamp. The interaction with the clamp is further stabilized by the exonuclease that is wedged between the two proteins and forms a second, indirect interaction between polymerase and clamp. Strikingly, a large conformational change in the polymerase switches its tail domain from interacting with the clamp in the DNA-bound structure, to more than 30 Å away from the clamp in the DNA-free structure. Finally, the processivity switch $\tau_c$ binds the tail of the polymerase and appears to sequester the polymerase tail away from the clamp in the DNA-free structure. Hence, our structures provide crucial insights into the regulation of the replicative DNA polymerase PolIII$\alpha$ by its associated proteins clamp, exonuclease and $\tau_c$. They furthermore form a crucial step towards determining the structure of the complete DNA polymerase III holoenzyme.

## Results

### Structure determination of the PolIIIα-clamp-exonuclease-$\tau_{500}$ complex

The interaction between PolIIIα and the clamp is weak, in the order of 1 μM (*Toste Rêgo et al., 2013*), and is not sufficient to maintain an intact complex at the low concentrations used for cryo-EM. Therefore, to stabilize the complex we altered the sequences of the clamp binding motifs of PolIIIα and the exonuclease to increase the affinity for the clamp. For this we used sequences derived from the translesion DNA polymerase UmuC and the DNA replication initiation factor Hda that out of a panel of 15 peptide sequences were the most potent inhibitors of the interaction between the polymerase and clamp (*Wijffels et al., 2004*) (see Materials and methods for more details). The obtained complex is >100 fold more stable than the wild-type complex (*Figure 1—figure supplement 1A*) This stabilized complex of PolIIIα, clamp and exonuclease was used together with $\tau_{500}$ (the polymerase-binding domain of τ: residues 500–643) and a 25 base pair (bp) DNA substrate to prepare samples for cryo-EM (*Figure 1—figure supplement 2A,B*). Three structurally distinct groups of particles could be identified from a single data set (63,215 particles). Two of these represent the PolIIIα-clamp-exonuclease-$\tau_{500}$ with and without DNA bound (*Figure 1*, *Videos 1* and *2*). The third class contains DNA too, but in this complex the tail domain of the polymerase and $\tau_{500}$ are not visible due to structural heterogeneity. The DNA-bound (5663 particles) and DNA-free (16,970 particles) structures were refined to 8.0 and 8.3 Å resolution, respectively (see *Figure 1—figure supplement 2* for details). The remaining particles (40,582) were classified into the third class in which the tail domain is not visible. Due to the larger number of particles, this structure was refined to 7.3 Å resolution. As this structure is otherwise identical to the complete DNA-bound complex, it will not be discussed further.

### Overall structure of the complex

The cryo-EM maps enable the unambiguous fitting of the high-resolution structures of the individual subunitsinto the cryo-EM maps (*Figure 1B,C*). No conformational changes were required for the fitting of the clamp, exonuclease or $\tau_{500}$, while the polymerase was divided into five domains that were independently fitted into density as rigid bodies (see *Figure 1—figure supplement 3*). None of the loops were modified, with the exception of the clamp binding motifs of the polymerase and exonuclease that were modeled after existing crystal structures of clamp-bound peptides from Pol II and Pol IV (*Georgescu et al., 2008a*; *Bunting et al., 2003*). B-form DNA was used for the DNA substrate, except for the last four base pairs that deviate from B-form DNA and were modeled after the DNA substrate from the *Thermus aquaticus* (Taq) PolIIIα crystal structure (*Wing et al., 2008*).

We describe the DNA-free complex first (*Figure 2*). The overall conformation of PolIIIα resembles that of the X-ray structure of *E. coli* and Taq PolIIIα (*Lamers et al., 2006*; *Bailey et al., 2006*) and reveals only a ~15˚ rotation of the fingers domain between the two structures (*Figure 1—figure supplement 3*). PolIIIα interacts with the clamp through the internal clamp binding motif (residues 920–924) (*Dohrmann and McHenry, 2005*; *Toste Rêgo et al., 2013*) that binds in the canonical binding pocket of the clamp (*Figure 2B*). Immediately after the clamp binding motif the density for the polymerase disappears, and resumes ~10 residues later, just before the oligonucleotide/oligosaccharide binding (OB) domain, indicating that this region of the polymerase is flexible (*Figure 2A*, left and middle panel).

On the other side of the complex, across the opening of the clamp, the PHP domain of the polymerase comes close to, but makes no contacts with the clamp (*Figure 2A*, left panel). Instead, the exonuclease is wedged between the clamp and the thumb domain of PolIIIα (*Figure 2A*, right panel). The catalytic domain of the exonuclease is in direct contact with the polymerase thumb domain whereas the contact with the clamp is mediated via a canonical clamp binding motif that is located immediately downstream of the catalytic domain (*Toste Rêgo et al., 2013*; *Jergic et al., 2013*). This clamp binding motif is bound to the pocket of the clamp in a manner similar to the polymerase in the other half of the clamp (*cf. Figure 2B,C*) and hence both pockets of the dimeric clamp are occupied in the ternary complex. Downstream of the clamp binding motif the tail of the exonuclease follows the contours of the polymerase PHP domain, where it is disordered for a stretch of ~15 residues that were shown to be mobile by NMR (*Ozawa et al., 2013*). Finally, the C-terminal helix of the exonuclease that mediates most of the binding to the polymerase (*Ozawa et al., 2013*)

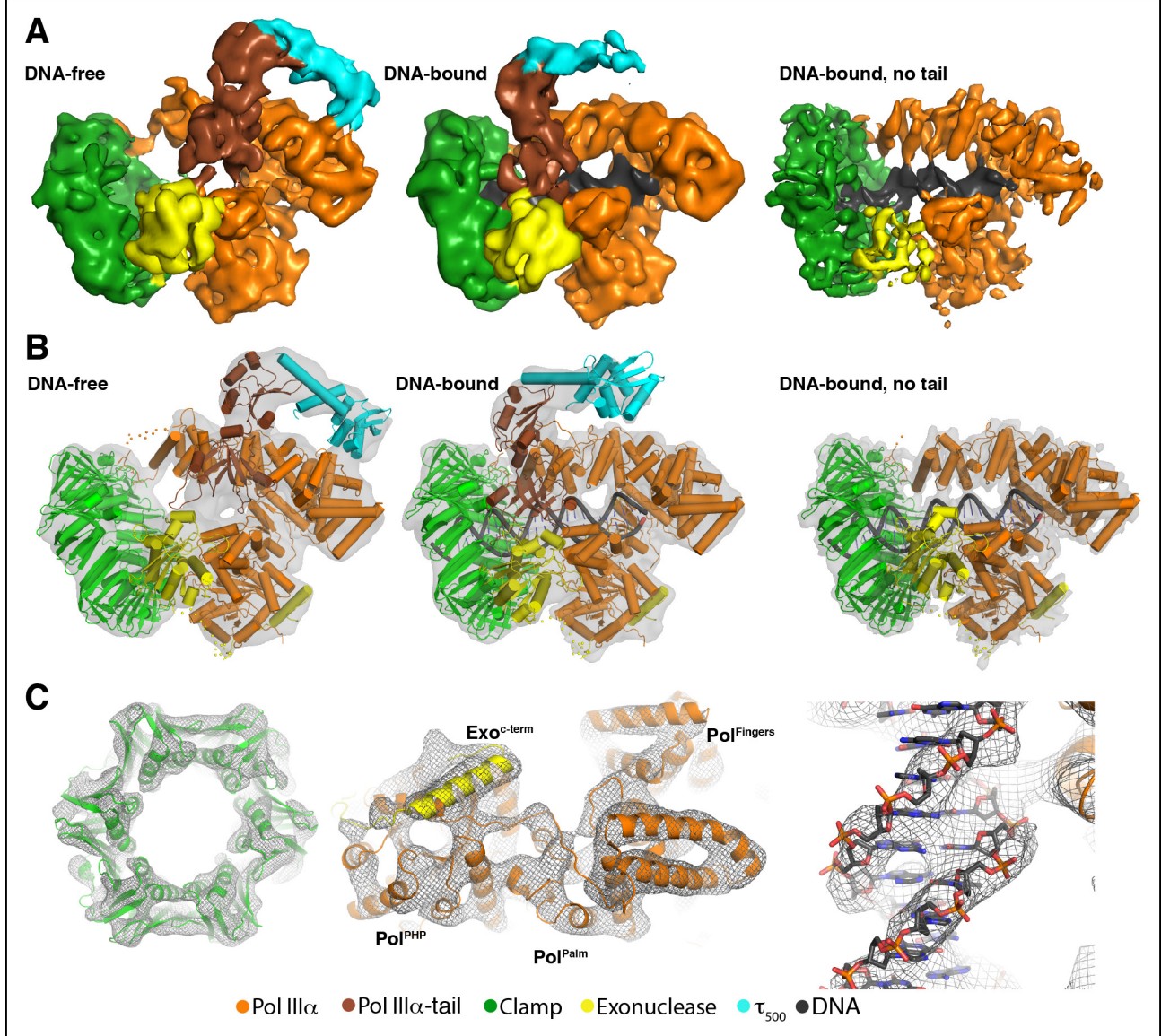

**Figure 1.** Cryo-EM structures of the *E. coli* PolIIIα-clamp-exonuclease-τ$_{500}$ complex. (A) Surface representation of the three structures, shown at 5 σ. Left to right: DNA-free, DNA-bound, and DNA-bound without tail. Colors indicate the position of the different proteins (B) Individual structures of PolIIIα, clamp, exonuclease, and τ$_{500}$ fitted into the cryo-EM map (shown in grey at 5 σ) (C) Detailed views of the cryo-EM map (shown in grey mesh at 6 σ). Left panel: exit channel of the clamp in the DNA-free structure showing the 'DNA-free' map. Middle panel: bottom view of the polymerase showing the 'DNA-free' map. Right panel: detail of the DNA showing the 'DNA-bound, no tail' map. See also *Videos 1 and 2*.

The following figure supplements are available for Figure 1:

**Figure supplement 1.** Characterization of improved clamp binding mutants.

**Figure supplement 2.** Microscopy data analysis and validation.

**Figure supplement 3.** Rigid body movements in PolIIIα.

packs tightly against the PHP domain of PolIIIα (*Figure 2A*, left and right panel), similar to the crystal structure of the PolIIIα-PHP domain and C-terminal helix of the exonuclease (*Ozawa et al., 2013*). Hence, in the ternary complex the exonuclease simultaneously binds the polymerase and clamp. By doing so it strengthens the association between the two proteins (*Toste Rêgo et al., 2013*), which is

required for processive DNA synthesis (*Jergic et al., 2013*). Previously, we built an approximate model for the polymerase, clamp and exonuclease complex, using distance restraints provided by chemical cross-linking coupled to mass-spectrometry (*Toste Rêgo et al., 2013*). When we map the same cross-links onto the cryo-EM model we find an improved fit of the cross-links with the model, due to the conformational changes in the complex as well as the detailed information about the interactions between the proteins that could not be modeled based on the cross-links alone (*Figure 2—figure supplement 1*).

The NMR structure of residues 500–621 of τ (*Su et al., 2007*) can be fitted into the density between the tail and fingers domain of the polymerase (*Figure 2D* and *Figure 2—figure supplement 2A*). The C-terminal end of τ is in contact with the tail of the polymerase. Unfortunately, the last 23 residues of τ that bind the polymerase (*Jergic et al., 2007*) are not part of the NMR structure and are therefore not present in our model. A second contact is found between the globular domain of $\tau_{500}$ and the fingers domain of the polymerase. This contact is mediated by residues 530–535 and 562–566 of $\tau_{500}$ and residues 657–667 of PolIIIα (*Figure 2—figure supplement 2B*). The position of $\tau_{500}$ in this complex is different from the position of the C-terminal domain of τ in complex with Taq PolIIIα, where it is located at the opposite side of the polymerase tail (*Figure 2—figure supplement 3*) (*Liu et al., 2013*). It must be noted though that the C-terminal domain of τ from *E. coli* and Taq share no sequence or structural homology and therefore engage with the polymerase in different ways.

## DNA binding in the PolIIIα-clamp-exonuclease-$\tau_{500}$ complex

In the DNA-bound complex (*Figure 3*), the entire length of the 25 base pair duplex is in contact with protein (*Figure 3A*). The position of the DNA is similar to that of the DNA in the crystal structure of Taq PolIIIα and *Geobacillus kaustophilus* PolC (*Wing et al., 2008*; *Evans et al., 2008*) (*Figure 3—figure supplement 1*). No density is observed for the 4 nucleotide (nt) single stranded overhang on the template strand indicating that this part of the DNA is flexible. In the complex, all contacts to the DNA are mediated by the thumb, palm and fingers domains of the polymerase and the inner surface of the clamp. No contacts to the DNA are made by the polymerase OB domain, the exonuclease, or $\tau_{500}$. The most extensive DNA contacts occur at the primer 3' end in polymerase active site where the thumb, palm and fingers domain of the polymerase contact the first 9 base pairs of the DNA duplex. It is also here that the only non-backbone contact is made by a loop of the thumb (residues 464–470), which is inserted into the major groove of the DNA (*Figure 3B*).

Away from the active site, the tip of the fingers domain (i.e. little finger [*Lamers et al., 2006*] or β binding domain [*Bailey et al., 2006*]), makes additional contacts to the DNA. Here, several positively charged residues (K831, K872, R876, R877, K881) as well as the positive charge of the helix dipole of

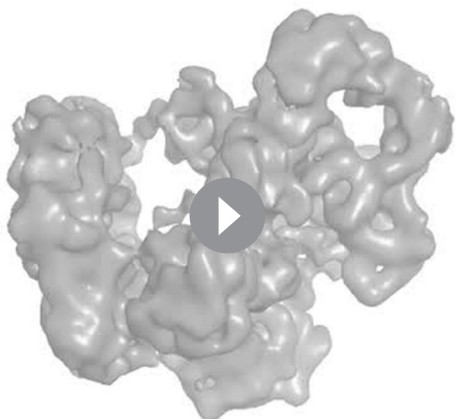 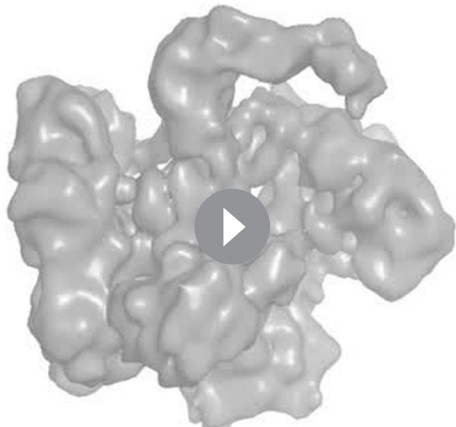

**Video 1.** Structure of the DNA-free complex of PolIIα-clamp-exonuclease-$\tau_{500}$, Related to *Figure 1*. Fitting of the high-resolution structures into the cryo-EM map of the DNA-free complex.

**Video 2.** Structure of the DNA-bound complex of PolIIα-clamp-exonuclease-$\tau_{500}$, Related to *Figure 1*. Fitting of the high-resolution structures into the cryo-EM map of the DNA-bound complex.

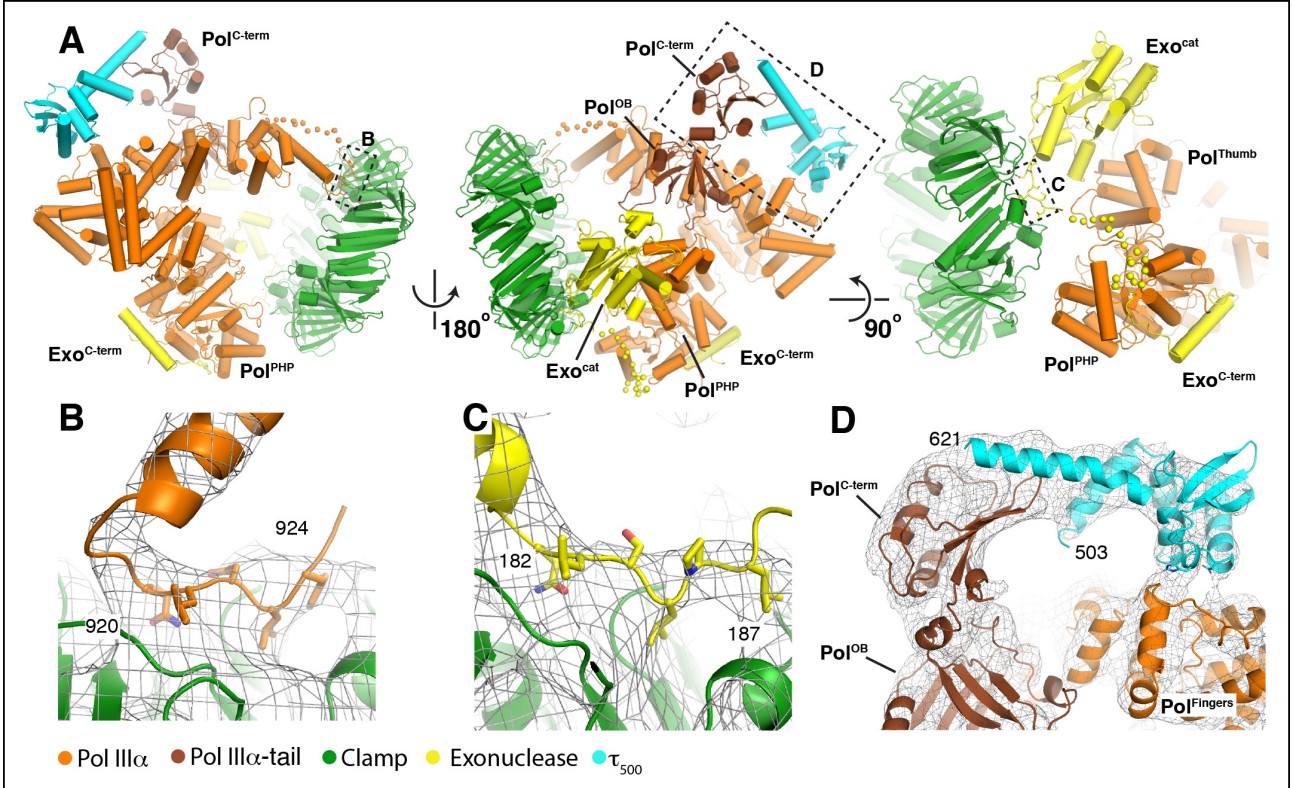

**Figure 2.** Multiple contacts between the subunits hold the complex together. (**A**) Three different views of the DNA-free complex of PolIIIα-clamp-exonuclease-τ500 showing extensive contacts between the polymerase and other subunits. Missing loops in PolIIIα (residues 927–936) and exonuclease (residues 190–207) are shown in dots. Dashed boxes indicate views shown in panels B-D. (**B**) Modified clamp binding motif of PolIIIα (QLDLF: shown in sticks) modeled into the binding pocket of the clamp. (**C**) Modified clamp binding motif of the exonuclease (QLSLPL: shown in sticks) modeled into the second binding pocket of the dimeric clamp. (**D**) τ500 simultaneously binds the fingers and tail domain of the polymerase. The C-terminal residues of τ500 (residues 622–643: not modeled) bind to the tail of the polymerase, while the globular domain of τ500 binds to the polymerase fingers domain (see *Figure 2—figure supplement 2* for more details).

The following figure supplements are available for Figure 2:

**Figure supplement 1.** Previously determined cross-links fit accurately with the cryo-EM model.

**Figure supplement 2.** Details of the interactions between τ500 and the PolIIIα fingers domain.

**Figure supplement 3.** Comparison of τ binding in *E. coli* and Taq PolIIIα.

two helices (residues 842–856 and 875–886) are pointed towards the DNA backbone (*Figure 3A*). At this position, the OB domain is in close proximity of the DNA but makes no contacts with it (*Figure 3C*). Instead, the OB domain forms a bridge between the PolIIIα fingers domain, thumb domain, and the exonuclease. Furthermore, while the isolated OB domain has been shown to bind to ssDNA (*Georgescu et al., 2009*), in this complex the OB domain is ~40 Å away from the ssDNA template. The DNA furthermore interacts with the clamp that surrounds the DNA like a nut around a bolt (*Figure 3D*). Several non-specific contacts are made to the backbone of the DNA providing an electrostatic cushion for the DNA to pass through as it leaves the complex. Compared to the crystal structure of the isolated clamp bound to DNA (*Georgescu et al., 2008a*) the clamp is rotated by ~20°, resulting in an almost perpendicular orientation (~80°) with respect to the DNA. In the 19 Å, negative stain EM structure of *Pyrococcus furiosus* PolB, the only other known structure of a DNA polymerase in complex with clamp and DNA, the DNA runs straight through the clamp as well (*Mayanagi et al., 2011*).

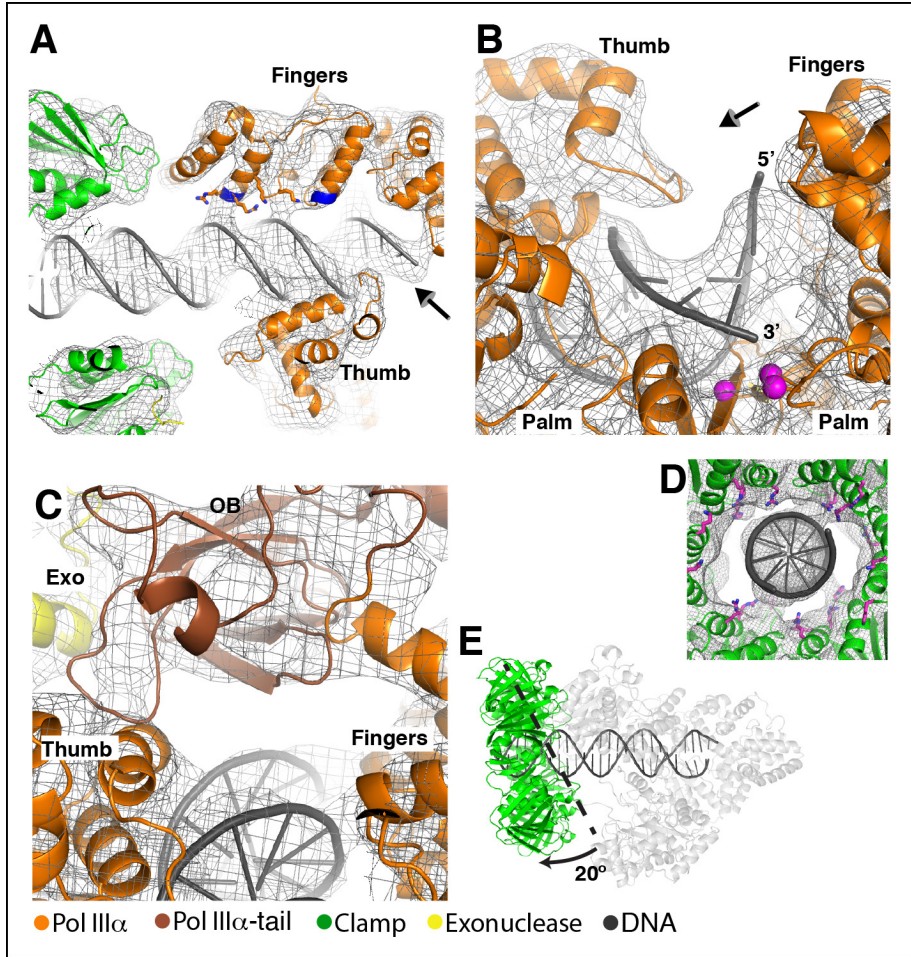

**Figure 3.** The DNA has extensive contacts with PolIIIα and clamp. (**A**) Overview of the DNA-bound complex. The N-termini of the two helices that point at the DNA backbone are colored in blue. Potential DNA interacting side chains are shown in sticks. The tail of PolIIIα, the exonuclease and τ$_{500}$ are omitted for clarity. Arrow indicates viewpoint in panel B (**B**) Polymerase active site, with the DNA held between thumb, palm and fingers domain. Polymerase active site residues are indicated with magenta spheres. Arrow indicates viewpoint in panel C (**C**) DNA interactions downstream of the active site. The OB domain is positioned on top of the DNA but does not make any contacts with it. (**D**) DNA exit channel in the clamp with positively charged residues within 10 Å of the DNA indicated in magenta sticks. *Note that the positions of the side chains have not been refined and should be seen as approximate positions.* (**E**) In the DNA-bound complex, the clamp is at a ~80° angle from the DNA. A dashed line indicates the position of the clamp alone bound to a DNA substrate. (***Georgescu et al., 2008a***). The other subunits (PolIIIα, exonuclease, τ$_{500}$) are shown in light grey for clarity.

The following figure supplements are available for Figure 3:

**Figure supplement 1.** Comparison of DNA binding by C family DNA polymerases.

**Figure supplement 2.** Pol IIIα has more extensive DNA interactions than other DNA polymerases.

PolIIIα is an extremely fast DNA polymerase with DNA synthesis speeds of up to 600–1000 nt/s (***Mok and Marians, 1987***; ***Mcinerney et al., 2007***). In contrast, the *E. coli* DNA polymerases Pol I, Pol II, and Pol IV have synthesis speeds of 15, 10, and 1 nt/s, respectively (***Schwartz and Quake, 2009***; ***Indiani et al., 2009***). Furthermore, in isolation PolIIIα has a surprisingly low affinity for DNA when compared to the other E. coli DNA polymerase (***Figure 3—figure supplement 2A***). Because of its weak binding to DNA, PolIIIα must therefore have developed a different way to keep itself correctly positioned on the DNA during rapid DNA synthesis. To this effect, PolIIIα may have evolved

its uniquely long fingers domain that is more than twice as long as the other *E. coli* DNA polymerases (*Figure 3—figure supplement 2B*). These polymerases have considerably smaller fingers domains, use shorter regions of DNA contacts, and have shorter predicted distances between the polymerase active site and the clamp (*Figure 3—figure supplement 2B*). In PolIIIα, the number of base pairs between the 3' end of the primer and the opening of the clamp is 22, while the predicted number of base pairs for Pol I, Pol II, and Pol IV is ~15. This unusually long DNA-protein contact appears to be well suited to accurately position the DNA without requiring tight binding that could slow down the translocation of the DNA. At the same time, the sequence-independent backbone contacts and the perfectly straight B-form DNA may facilitate the rapid exit of the DNA from the active site. The active site itself is wrapped tightly around the DNA where PolIIIα is the only polymerase that inserts a loop of the thumb domain into the major groove of the DNA, while the thumb domains of Pol I, Pol II, and Pol IV only have backbone contacts with the DNA (*Figure 3—figure supplement 2C*). Hence, it seems plausible that this combination of unique contacts with the DNA may have evolved to support the high speeds of DNA synthesis by PolIIIα without compromising accuracy.

## DNA binding induces a large conformational change in the complex

To enable the many contacts with the DNA, the complex undergoes extensive conformational changes from the DNA-free to the DNA-bound state. Most prominent is a ~35° rotation of the polymerase tail and $\tau_{500}$, which move from a position over the polymerase active site to a position adjacent to the sliding clamp (*Figures 1,4* and *Video 3*). The simultaneous movement of the polymerase tail and $\tau_c$ results in a 70 Å displacement of the globular domain of $\tau_{500}$. The tail of PolIIIα consists of the OB domain (residues 960–1071) and the C-terminal τ-binding domain (residues 1079–1160). Together with $\tau_{500}$ they form a rigid structure that shows few changes between the DNA free and DNA bound structure, indicating that the interaction between $\tau_{500}$ and the tail of PolIIIα must be stable (*Figure 4C*). As a result of the repositioning of the polymerase tail, the contact between $\tau_{500}$ and the fingers domain of the polymerase is broken and a new contact between the OB domain and the clamp is forged (*Figure 4B* and *Video 3*). The OB domain makes two new contacts with the clamp via a short helix (1035–1043) and a long protruding loop (1003–1013) (*Figure 4D,E*). These motifs contact the clamp at loops 24–28 and 275–278, respectively. Hence in the DNA-bound complex, the polymerase has three points of contact to the clamp: one via the canonical clamp binding motif (residues 920–925: *Figure 2B*); one indirectly via the exonuclease (*Figure 2C*); and one contact via the OB domain (*Figure 4D,E*). Previously, it has been shown that a triple mutation in OB domain result in reduced DNA synthesis (*Georgescu et al., 2009*) which was attributed to the loss of ssDNA binding by the OB domain. However, in our structure the OB domain is far away (~40 Å) from the ssDNA overhang of the template strand. Instead, the mutations (R1004S, K1009S, R1010S) are located at the interface between the OB domain and the clamp (*Figure 4E*) and therefore could weaken the interaction between the polymerase and clamp, providing an alternative explanation for the reduced DNA synthesis.

## Discussion

The *E. coli* replisome consists of 12 different proteins that can be divided into three subcomplexes: the helicase-primase complex, the clamp loader complex, and the PolIIIα-clamp-exonuclease-$\tau_c$ complex. The structures of two of the three subcomplexes have been determined previously: the helicase-primase complex (*Bailey et al., 2007*; *Wang et al., 2008*), and the clamp loader complex (*Jeruzalmi et al., 2001*; *Simonetta et al., 2009*). The structure of the PolIIIα-clamp-exonuclease-$\tau_{500}$ complex on the other hand has remained elusive due to its dynamic nature. The cryo-EM structures of the PolIIIα-clamp-exonuclease-$\tau_{500}$ complex presented in this work finally reveal the nature of the interactions in the ternary complex and are a crucial step forward towards determining the structure of the complete bacterial replisome.

Our cryo-EM structures furthermore provide a crucial insight into the structural organization of the replicative DNA polymerase and its associated proteins clamp, exonuclease and $\tau_{500}$. They show how the clamp and exonuclease tether the polymerase to the DNA through multiple contacts. Importantly, they also reveal a large conformational change where the tail of the polymerase moves from interacting with the clamp in the DNA-bound state, to a position 35 Å away from the clamp in

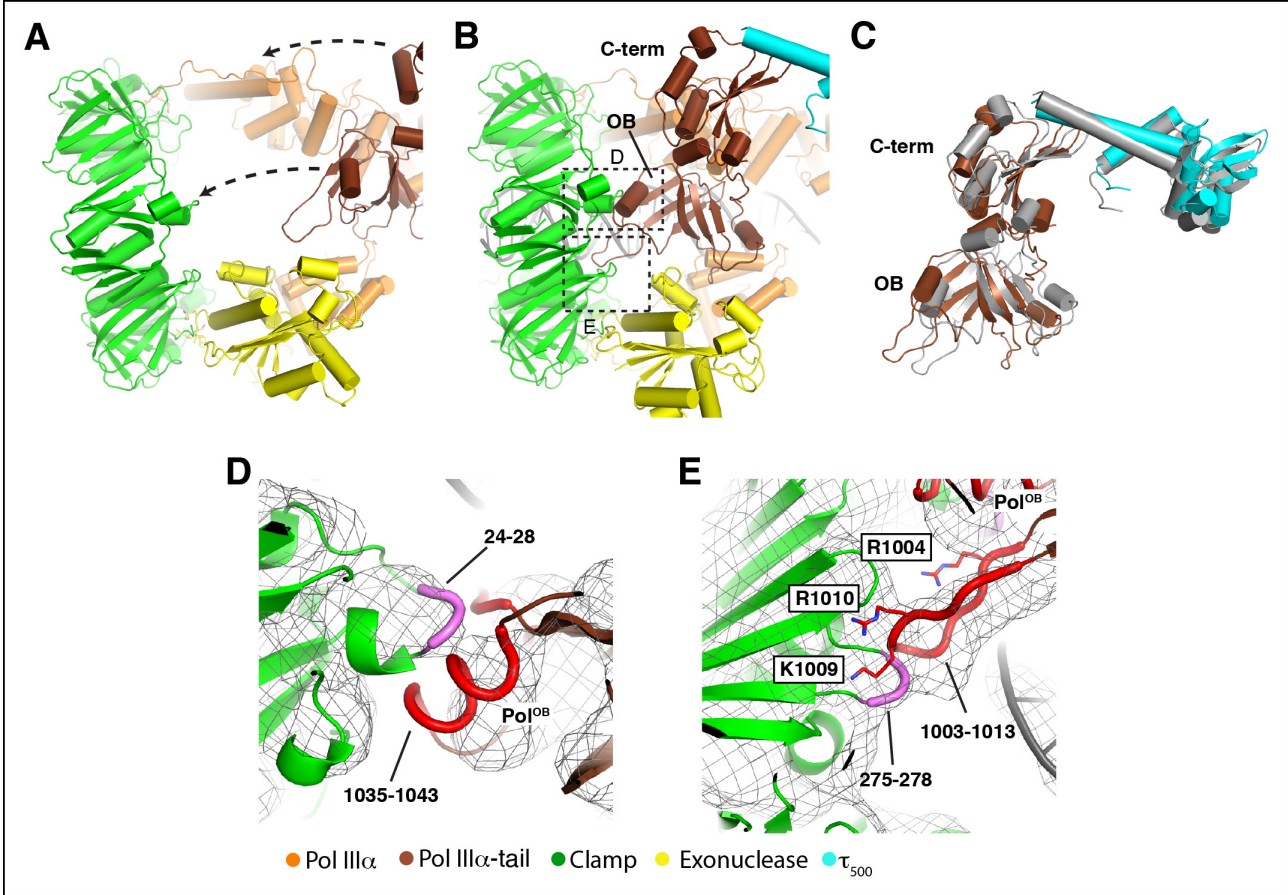

**Figure 4.** DNA binding induces large conformational changes in the polymerase. (**A**) Clamp binding by PolIIIα in the DNA-free complex. Arrows indicate movement of the PolIIIα tail (see also **Video 3**). (**B**) Clamp binding by PolIIIα in the DNA-bound complex. Dashed boxes indicate views shown in panel D and E (**C**) Comparison of the PolIIIα-tail - $\tau_{500}$ interaction in the DNA-free (in grey) and DNA-bound structure. (**D and E**) Detailed view of the clamp - PolIIIα OB domain interaction. Interacting regions at the interface are indicated in thick coil in magenta (clamp: residues 24–24 and residues 275–278) and red (PolIIIα-OB domain: residues 1035–1043 and residues 1003–1013). Residues mutated in Georgescu *et al* (**Georgescu et al., 2009**) are shown in sticks and labeled with outlined boxes. *Note that the positions of the side chains have not been refined and should be seen as approximate positions.*

the DNA-free state. What may be the role for such a conformational change? On the lagging strand, the polymerase repositions to a newly primed site every ~1000 bp. To do so, the polymerase needs to release both clamp and DNA. We propose that the switch-like movement of the polymerase tail may play a part in the release and consequent repositioning of the polymerase at the end of the Okazaki fragment. A hypothetical model describing how this could work is presented in *Figure 5*. During DNA synthesis, the tail of the polymerase is bound to the clamp, stabilizing the interaction between polymerase and clamp (marked with '1' in *Figure 5*). This confirmation may be further stabilized by the presence of a DNA binding region immediately upstream of $\tau_{500}$ (marked with '2') (*Jergic et al., 2007*). Upon encounter of a release signal, $\tau_{500}$ rebind to the polymerase fingers domain (marked with '3') thus sequestering the polymerase tail away from the clamp (marked with '4') and initiating the dissociation of the polymerase from clamp and DNA. What could serve as the release trigger? Two non-exclusive models have been proposed (*Li and Marians, 2000*). In the 'collision' model, the encounter with the dsDNA of the previous Okazaki fragment induces the release of the polymerase. In support of this model, it has been shown that a decreasing gap size between the 3' terminus of the lagging strand and the 5' end of the previously synthesized Okazaki fragment promotes release of the polymerase (*Leu et al., 2003*; *Georgescu et al., 2009*; *Dohrmann et al., 2011*). Two possible sensors for the decreasing gap on the lagging strand have been suggested. The ssDNA binding properties of the OB domain in the tail of the polymerase has been proposed to

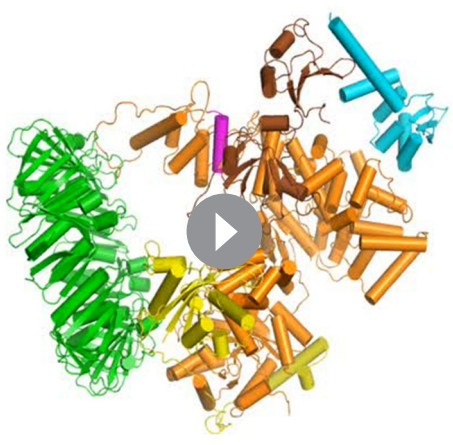

**Video 3.** DNA binding induces large conformational changes in the complex, Related to *Figure 4*. Linear morphing of the DNA-free to DNA-bound state showing the large conformational change between the two states.

play a role in the sensing of the ssDNA vs dsDNA (*Georgescu et al., 2009*). Yet our cryo-EM models show that the OB domain is ~40 Å away from the ssDNA and that it is involved in binding to the sliding clamp. Alternatively, it has been proposed that the C-terminal fragment of τ may act as the sensor as it is required to release the polymerase from a decreasing gap size (*Leu et al., 2003*). Indeed, it was found that the region in τ, immediately upstream of $\tau_{500}$, has DNA binding affinity (*Jergic et al., 2007*).

Contesting the collision model is the observation that the release of the polymerase by a decreasing gap size is too slow ($t_{1/2}$ = 110 s) for the frequency at which the lagging strand polymerase is re-positioned (every 1–2 s) (*Dohrmann et al., 2011*), suggesting that additional or alternative release factors are required. The alternative 'signaling' model therefore proposes that the trigger comes from one of the other components of the replisome such as the RNA primase DnaG, based on the observation that the increased primase concentration results

in shorter lagging strand fragments (*Wu, Zechner and Marians, 1992*; *Li and Marians, 2000*). However, it was recently shown that the presence of a primer alone is sufficient to induce release at the lagging strand and that activity of the primase is not required (*Yuan and McHenry, 2014*). How the presence of the RNA primer is signaled to the polymerase remains unclear. Yet, the fact that the τ protein is both part of the clamp loader complex that positions clamps onto the primer and simultaneously binds the polymerase makes this a suitable conveyor of the signal. While our structures do not discriminate between the type of release trigger for the lagging strand polymerase, they do now provide the means to test the precise workings of the molecular switch that enables the release of the polymerase.

## Materials and methods

### Materials
All chemicals and oligonucleotides were purchased from Sigma-Aldrich (Gillingham, United Kingdom) and chromatography columns from GE healthcare (Little Chalfont, United Kingdom).

### Protein expression and purification
To increase binding to the clamp, amino acid residues 920–924 of *E. coli* PolIIIα were changed by site directed mutagenesis from QADMF to QLDLF, while in the exonuclease residues 182–187 were changed from QTSMAF to QLSLPL, based on sequences described in (*Wijffels et al., 2004*). All proteins were expressed in *E. coli* (DE3) BL21. PolIIIα, clamp and exonuclease were expressed and purified as described before (*Toste Rêgo et al., 2013*). $\tau_{500}$ was purified by Histrap HP column, Resource S column, and a Superdex 75 gel filtration column. His-tags were removed by proteolytic cleavage with human rhinovirus 3C protease. Proteins were flash frozen in liquid nitrogen and stored at -80°.

### Gel filtration analysis
Proteins were analyzed by gel filtration using a 2.4 mL Superdex 200 Increase column (GE healthcare) in 25 mM Hepes pH 7.5, 150 mM NaCl, and 2 mM DTT. PolIIIα-clamp-exonuclease complex was assembled at 10, 1, and 0.1 μM and 50 μL injected onto the column.

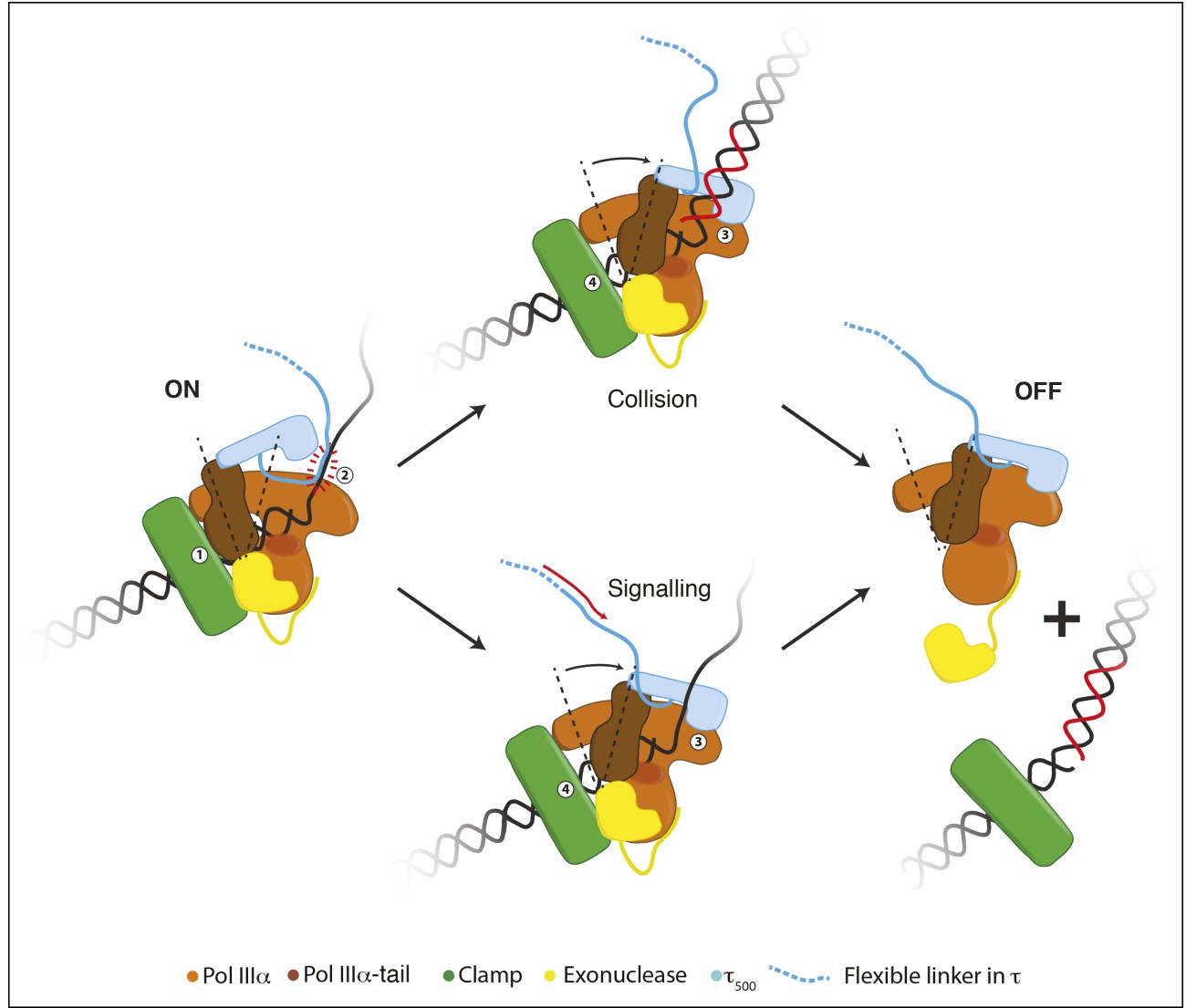

**Figure 5.** Schematic representation for a possible role of the conformational changes in the polymerase. During processive DNA synthesis, the tail of the polymerase is attached to the clamp (indicated with '1') and pulls $\tau_{500}$ away from the polymerase fingers domain. This conformation may be further stabilized by the presence of a DNA binding region immediately upstream of $\tau_{500}$ (indicated with '2'; see text for more details). Upon encounter of a release trigger, the contact between $\tau_{500}$ and the polymerase fingers domain is restored (indicated with '3'), and the contact between the clamp and polymerase tail is broken (indicated with '4'). The release trigger may either come from a collision with the previous Okazaki fragment (indicated with 'Collision'), or a signal from other replisome components via the flexible linker of $\tau$ (indicated with 'Signaling'). Once the polymerase-tail clamp contact has been broken, the two remaining contacts between the clamp and polymerase-exonuclease are not enough to keep the polymerase bound to the clamp. The polymerase is released from clamp and DNA and can be repositioned to a newly primed site to reinitiate DNA synthesis.

## Electro mobility shift assay

A DNA substrate identical to the substrate used for the cryo-EM samples was used, with the exception of a 6-carboxyfluorescein (6-FAM) at the 5′ end of the primer strand and a phosphorothioate link at the 3′ terminal bond to prevent exonuclease digestion. 5 nM DNA was incubated with 2.5 µM polymerase (*E. coli* Pol I (Klenow fragment), Pol II, Pol IIIα, or Pol IV) for 10 min at room temperature. Reaction mixtures contained 20 mM Tris pH 7.5, 4% glycerol, 5 mM DTT, 40 µg/ml BSA, and 40 mM Potassium Glutamate. Half of the sample was separated on a native 6% acrylamide gel and imaged on a Typhoon laser scanner (GE Healthcare). The remaining half of the sample was analyzed on a denaturing 4–12% SDS acrylamide gel and stained with Coomassie blue.

## Sample preparation for cryo-EM

The PolIIIα-clamp-exonuclease-$\tau_{500}$ protein complex was assembled from the individual components to a final concentration of 15 µM and purified on a 2.4 mL Superdex 200 Increase gel filtration column (GE Healthcare) in 25 mM Hepes pH 7.5, 50 mM Potassium Glutamate, 3 mM Magnesium Acetate, and 2 mM DTT. The peak fraction (~4 µM) was retrieved and incubated for 5 min with 20 µM of a 25 bp DNA substrate with a 4 nt overhang (template: 5ʹ-TCAGGAGTCCTTCGTCCTAGTACTAC-TCC-3ʹ, primer: 5ʹ-GGAGTAGTACTAGGACGAAGGACTC-3ʹ) for 5 min at room temperature. Subsequently, 0.1 volume of 0.05% (V/V) Tween 20 was added and incubated for another 5 min before the samples were pipetted onto glow-discharged holey carbon cryo-EM grids (Quantifoil Cu R1.2/1.3), and frozen in liquid ethane using a Vitrobot (FEI, Hillsboro, OR).

## Data collection and image processing

All data was collected using a Titan Krios electron microscope (FEI) operated at 300 kV equipped with a K2 summit direct electron detector (Gatan, Pleasaston, CA). Although this detector was mounted after a Gatan Imaging Filter (GIF), the filter was not used to remove any inelastic scattering. Images were collected in single-electron counting mode at a calibrated magnification of 28.571x (1.76 Å/pixel), using a flux of 2 e/Å²/sec and a total dose of 40 e/Å² over a total of 20 frames. Frames were aligned and averaged using whole-image movement correction using MOTIONCORR (*Li et al., 2013*). Contrast transfer function parameters were calculated using CTFFIND3 (*Mindell and Grigorieff, 2003*). All subsequent particle picking and data processing was performed using Relion-1.3 (*Scheres, 2012*), with the exception of the generation of the initial model, which was done using Eman2 (*Tang et al., 2007*). A total of 1350 micrographs were recorded from which >550,000 particles were picked automatically in Relion. After 2D classification, a large number of spurious particles as well as particles that show free polymerase or free clamp were removed, yielding a dataset of ~90,000 particles. After 3D classification a another ~27,000 were removed to yield a final dataset of 63,215 particles. From these, six 3D classes were calculated that were subsequently merged into the final three 3D classes of 'DNA-free' (16,970 particles), 'DNA-bound' (5663 particles) and 'DNA-bound, no tail' (40,582 particles). Particle-based movement correction and per-frame B-factor weighting to account for radiation damage and unresolved particle movement was performed in the later stages of refinement using the particle polishing option in Relion (*Scheres, 2014*). Reported resolutions are based on the gold-standard FSC-0.143 criterion (*Scheres and Chen, 2012*) and FSC-curves were corrected for the convolution effects of a soft mask using high-resolution noise-substitution (*Chen et al., 2013*). All density maps were sharpened by applying a negative B-factor that was estimated using automated procedures (*Rosenthal and Henderson, 2003*). We believe that the resolution of these reconstructions is limited by both the relatively small size of the complex (250 kDa), which hampers accurate alignment and classification, and the inherent flexibility of this four-protein and DNA complex. Still, the maps are of excellent quality, with individual helices, β-sheets, and loops clearly visible in the map (*Figure 1C*).

## Fitting of the crystal structures into the cryo-EM map

Individual crystal or NMR structures were manually placed into the cryo-EM map in PyMOL (*Schrödinger, LLC 2010*) and subsequently rigid-body fitted into the density using Coot (*Emsley et al., 2010*). PDB codes of the fitted structures are: PolIIIα: 2HNH (*Lamers et al., 2006*), clamp: 2POL (*Kong et al., 1992*), exonuclease: 1J54 (*Hamdan, et al., 2002*), $\tau_{500}$: 2AYA (*Su et al., 2007*). The C-terminal tail of Eco PolIIIα that is lacking in the crystal structure (2HNH) was modeled as described in (*Toste Rêgo et al., 2013*). The PolIIIα structure was divided into five domains that were further fitted independently into density as rigid bodies (see *Figure 1—figure supplement 3B, C*). These domains were: PHP (residues 1–280), palm-fingers (residues 281–432 + 510–810), thumb (residues 433–509), tip-of-fingers (residues 811–928) and C-terminal tail (residues 929–1160). Clamp binding motifs of PolIIIα and exonuclease were manually built into the clamp in Coot guided by the crystal structures of clamp-bound peptides from Pol II and Pol IV, (*Bunting et al., 2003*; *Georgescu et al., 2008b*; *Jeruzalmi et al., 2001*). The DNA substrate was generated with Coot, and the last four base pairs of the DNA were adjusted guided by the DNA from Taq Pol IIIα (*Wing et al., 2008*).

## Comparison of DNA polymerase structures

The following crystal structures of C family DNA polymerases were used to compare DNA binding and τ binding. DNA bound Taq PolIIIα (PDB code: 3E0D [*Wing et al., 2008*]), τ bound Taq PolIIIα (PDB code: 4IQJ [*Liu et al., 2013*]), DNA bound *G. kaustophilus* PolC (PDB code: 3F2B [*Evans et al., 2008*]). Crystal structures of bacterial DNA polymerases in complex with DNA were used to compare the distance between the polymerase active site and the opening to the clamp. The following structures were used: *T. aquaticus* DNA Pol I (PDB code: 1QTM [*Li et al., 1999*]), *E. coli* Pol II (PDB code: 3K57 [*Wang and Yang, 2009*]), *E. coli* PolIIIα (this work), and *E. coli* Pol IV (PDB code: 4IRD [*Sharma et al., 2013*]). For the structures of Pol I, Pol II, and Pol IV, the sliding clamp (PDB code: 2POL [*Kong et al., 1992*]) was manually placed close to the clamp binding sequences in the different polymerases, taking care not to cause any clashes with other parts of the polymerase.

## Acknowledgements

We thank members of the Lamers lab as well as Xiaochen Bai, Paula da Fonseca, and Nigel Unwin for useful suggestions. We also thank David Barford and Ana Toste-Rêgo for critically reading the manuscript. This work was supported by the UK Medical Research Council through grants U105197143 to MHL and MC_UP_A025_1013 to SHWS Cryo-EM density maps have been deposited with the Electron Microscopy Data Bank (accession numbers EMD-3198, EMD-3201, and EMD3202), and coordinates of the models have been deposited with the Protein Data Bank (PDB entry codes 5FKV, 5FKU, and 5FKW)

## Additional information

### Competing interests

SHWS: Reviewing editor, *eLife.* The other authors declare that no competing interests exist.

### Funding

| Funder | Grant reference number | Author |
|---|---|---|
| Medical Research Counc | U105197143 | Meindert H Lamers |
| Medical Research Council | MC_UP_A025_1013 | Sjors HW Scheres |

The funders had no role in study design, data collection and interpretation, or the decision to submit the work for publication.

### Author contributions

RFL, Conception and design, Acquisition of data, Analysis and interpretation of data, Drafting or revising the article; JC, Acquisition of data, Analysis and interpretation of data; SHWS, Analysis and interpretation of data, Drafting or revising the article; MHL, Conception and design, Analysis and interpretation of data, Drafting or revising the article, Contributed unpublished essential data or reagents

### Author ORCIDs

Rafael Fernandez-Leiro, http://orcid.org/0000-0002-7941-0357
Sjors HW Scheres, http://orcid.org/0000-0002-0462-6540

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
