## [Decision Letter]

Thank you for submitting your work entitled "cryo-EM structures of the *E.
coli* replicative DNA polymerase reveal dynamic interactions with clamp,
exonuclease and τ" for peer review at *eLife*. Your submission has
been favorably evaluated by Michael Marletta (Senior Editor), a Reviewing Editor, and
three reviewers. One of the three reviewers, Mike O’Donnell (Reviewer #3), has agreed to
reveal his identity.

The reviewers have discussed the reviews with one another and the Reviewing Editor has
drafted this decision to help you prepare a revised submission.

Using cryo-electron microscopy, Fernandez-Leiro et al. have determined the ~8 Å
resolution structures, in a DNA-bound and DNA-free state, of the holoenzyme catalytic
core comprising the *E. coli* DNA polymerase III α subunit, the DNA
sliding clamp β, the exonuclease ε, and the C-terminal domain of the clamp loader
subunit τ (τ_500_). Interesting findings include: 1) showing that the
rigid-body module composed of the C-terminal part of the clamp loader and the polymerase
tail undergoes a large rearrangement upon DNA binding, and 2) that there is a new
contact point between τ and the polymerase in the apo configuration, and different one,
formed in the presence of DNA, between the OB-fold domain of the polymerase and the
clamp. This work represents a highly significant advance in the science of sliding
clamps, and the mechanism by which DNA polymerase functions on them. This structural
study identifies an entirely new connection of the polymerase to the clamp. This new
connection only occurs when the polymerase binds a primed template. The connection is in
a new location, outside the two typical hydrophobic pockets on clamps that are the usual
target of protein trafficking.

In general, the paper is reasonably well-written and its organization is easy to follow.
The quality of the EM reconstructions is very good, especially considering the small
size of the complex (~250 kDa), and the heterogeneity present in the sample. The newly
identified contacts between the polymerase, clamp, and τ provide a logical and
long-desired molecular picture of why holoenzyme formation leads to improved
processivity and potentially of how PolIII alters its structural state in response to
binding DNA (which may be important for polymerase recycling during lagging strand
synthesis). Overall, the work provides valuable new insights into important DNA
replication events and stands to have a long-lasting impact on the field.

Before publication, the authors need to make the following essential revisions:

1) The authors refer to the study of Georgescu et al. (EMBO, 2009) and suggest that the
observed clamp-OB CTD interaction provides "an alternative explanation" for
the reduced DNA synthesis seen by them. However, the Discussion section remains
relatively vague and does not discuss the opposing model, although it is directly
relevant for Okazaki fragment sensing. While the entire mechanism of such sensing is
probably beyond the experimental scope of the present manuscript, the authors need to
provide a more considered discussion of "collision" models versus
"signaling". Specifically, the authors need to specifically address the
conclusions in Dohrmann et al. 2011, and related work.

2) Lamers and colleagues identified recently domain-domain interactions within
PolIII-clamp-exonuclease complex via cross-linking / mass spectrometry analysis (Toste
Rêgo et al. EMBO 2013). In the present manuscript they frequently refer to this study in
order to support their conclusions. Please show whether the crosslinking data is
consistent with the molecular models derived from cryo-EM by measuring respective
distances. Are there many outliers that exceed the length of the crosslinking reagent
and might indicate different conformations?

3) The manuscript needs a final figure that proposes the obvious function of this new
connection in the polymerase recycling process: a summary "mechanism" figure.
This figure should summarize the different movements/states, and illustrate the authors'
speculation as to how these changes might be involved in polymerase cycling during
lagging strand synthesis. An expert in the field will connect the dots without a model
figure, but for many readers, this finding deserves a figure and sufficient
explanation/speculation in the text, to convey the significance of this to
non-experts.

---

## [Author Response]

1) The authors refer to the study of Georgescu et al. (EMBO, 2009) and suggest
that the observed clamp-OB CTD interaction provides "an alternative
explanation" for the reduced DNA synthesis seen by them. However, the Discussion
section remains relatively vague and does not discuss the opposing model, although it
is directly relevant for Okazaki fragment sensing. While the entire mechanism of such
sensing is probably beyond the experimental scope of the present manuscript, the
authors need to provide a more considered discussion of "collision" models
versus "signaling". Specifically, the authors need to specifically address
the conclusions in Dohrmannet al. 2011, and related work.

We have now included a more extensive discussion regarding the "collision" vs.
"signaling" model. This includes the work by Dohrmann et al. 2011 and related
work.

2) Lamers and colleagues identified recently domain-domain interactions within
PolIII-clamp-exonuclease complex via cross-linking / mass spectrometry analysis
(Toste Rêgo et al. EMBO 2013). In the present manuscript they frequently refer to
this study in order to support their conclusions. Please show whether the
crosslinking data is consistent with the molecular models derived from cryo-EM by
measuring respective distances. Are there many outliers that exceed the length of the
crosslinking reagent and might indicate different conformations?

Indeed, our previous cross-linking results are entirely consistent with our cryo-EM
models. We now refer to this in the text, and have included an additional figure (Figure 2—figure supplement 1) showing the
cross-links mapped onto the old model (left) and the DNA-free cryo-EM model (right).

3) The manuscript needs a final figure that proposes the obvious function of
this new connection in the polymerase recycling process: a summary
"mechanism" figure. This figure should summarize the different
movements/states, and illustrate the authors' speculation as to how these changes
might be involved in polymerase cycling during lagging strand synthesis. An expert in
the field will connect the dots without a model figure, but for many readers, this
finding deserves a figure and sufficient explanation/speculation in the text, to
convey the significance of this to non-experts.

We have now added a final figure that summarizes in a schematic manner the possible role
of the conformational change in the polymerase. This figure is further explained in the
extended discussion about the "collision" vs. "signaling" model.